# Biomass Sorghum (*Sorghum bicolor*) Agronomic Response to *Melanaphis sorghi* (Hemiptera: Aphididae) Infestation and Silicon Application

**DOI:** 10.3390/insects16060566

**Published:** 2025-05-27

**Authors:** Douglas G. Santos, Leonardo L. C. Dias, Guilherme S. Avellar, Maria Lúcia F. Simeone, Rafael A. C. Parrella, Nathan M. Santos, Thaís F. Silva, Antônio A. Neto, Simone M. Mendes

**Affiliations:** 1Department of Biosystems Engineering, Federal University of São João del Rei (UFSJ), São João del-Rei 36307-352, Brazil; gracieldouglas@gmail.com (D.G.S.); guilherme_avellar@hotmail.com (G.S.A.); 2Department of Exact and Biological Sciences, Federal University of São João del-Rei (UFSJ), Sete Lagoas 35702-031, Brazil; leodias@ufsj.edu.br (L.L.C.D.); thaisfernandaagro@gmail.com (T.F.S.); antonioantunes_ufsj@hotmail.com (A.A.N.); 3Brazilian Agricultural Research Corporation (Embrapa), Sete Lagoas 35702-098, Brazil; marialucia.simeone@embrapa.br (M.L.F.S.); rafael.parrella@embrapa.br (R.A.C.P.); 4Department of Entomology, Federal University of Lavras (UFLA), Lavras 37203-202, Brazil; nathansantos741@gmail.com

**Keywords:** sorghum aphid, silicic acid, bioenergy crops, aphid management, pest resistance, plant biomass, silicon fertilization

## Abstract

This study investigated whether soil-applied silicon could enhance the resistance of biomass sorghum (BRS 716) (*Sorghum bicolor*) plants to aphid infestation (*Melanaphis sorghi* (Theobald, 1904) (Hemiptera: Aphididae)) while also boosting crop productivity. Sorghum plants were grown under controlled conditions and treated with varying silicon doses (0, 2, 4, and 6 tons per hectare). When exposed to aphid attacks, plants that received silicon displayed increased tolerance compared with untreated ones. Notably, the highest application rate (6 tons per hectare) was most effective, enabling plants to sustain healthy growth and generate greater biomass despite pest pressure. In addition to reinforcing resistance, silicon positively influenced key plant traits. It increased the amount of plant fiber (cellulose) and improved the uptake of essential nutrients such as phosphorus and calcium. However, it also led to a slight reduction in the concentration of certain other nutrients. These results highlight silicon’s multifaceted role in plant defense and development. Overall, this research supports the potential of silicon supplementation as part of an integrated pest and crop management strategy. Incorporating silicon into agricultural practices could contribute to more resilient and productive cropping systems, offering a promising alternative to conventional chemical controls and helping to promote sustainable farming methods.

## 1. Introduction

The transition to renewable and sustainable energy sources has become a global priority to address climate change and reduce dependence on fossil fuels [1,2]. In this context, biomass sorghum (*Sorghum bicolor*) emerges as a promising feedstock for bioenergy due to its high lignocellulosic biomass yield, rapid growth, and adaptability to diverse climates [3,4,5]. Energy cogeneration from sorghum combines thermochemical and biochemical processes [4]. This integrated approach efficiently converts biomass into thermal and electrical energy while reducing greenhouse gas emissions [6,7].

Despite its potential, sorghum productivity is severely compromised by infestations of the sorghum aphid, *Melanaphis sorghi* (Theobald, 1904) (Hemiptera: Aphididae) [8]. This invasive pest was initially identified in the Americas in 2013 [9]. It infests various *Sorghum* species, notably Johnson grass (*S. halepense*) and grain sorghum (*S. bicolor*), resulting in substantial yield reductions [10,11,12]. In Brazil, *M. sorghi* has been documented in several agricultural regions, and infestations by this invasive pest have resulted in losses of up to 20% of areas cultivated with sorghum [13].

The aphid feeds on phloem sap, leading to nutrient depletion that induces chlorosis, wilting, and plant necrosis [14]. Additionally, the honeydew excreted by *M. sorghi* promotes sooty mold growth, which coats leaf surfaces and impairs photosynthesis [14]. Current management strategies, predominantly reliant on chemical insecticides, face challenges such as pest resistance and environmental concerns [15], underscoring the need for sustainable alternatives.

Silicon (Si) supplementation has emerged as an eco-friendly approach to enhance plant resistance to biotic stresses [16]. While not essential for plant development, Si accumulates in plant tissues, reinforcing cell walls and influencing physiological responses to pests and nutrient imbalances [17]. In sorghum, a species known for its high Si uptake, this element contributes to greater structural integrity and stress tolerance [18,19].

Notably, Si reduces susceptibility to aphids in crops. In sorghum, Si application decreased feeding by the greenbug (*Schizaphis graminum* (Rondani, 1852) (Hemiptera: Aphididae)) [20], while high Si doses (>4 t ha^−1^) suppressed *M. sorghi*’s biological parameters in grain sorghum [21]. Similar effects were observed in wheat, where Si activated jasmonic acid pathways [22] and enhanced resistance to *Sitobion avenae* (Fabricius, 1775) (Hemiptera: Aphididae) and *Rhopalosiphum padi* (Linnaeus, 1758) (Hemiptera: Aphididae) [23,24,25]. However, the mechanisms underlying Si-induced resistance in biomass sorghum under *M. sorghi* infestation remain underexplored, particularly its effects on agronomic traits and biomass quality.

This study evaluates the role of Si in enhancing the resistance of biomass sorghum (hybrid BRS716) to *M. sorghi* and its impact on crop productivity. We hypothesize that Si doses (2–6 t ha^−1^) will (1) reduce aphid infestation and damage and (2) improve agronomic performance and biomass composition. The null hypothesis posits no differential effects across Si doses (0–6 t ha^−1^) under infestation. Our findings aim to advance integrated pest management strategies for sustainable bioenergy production.

## 2. Materials and Methods

This experiment was conducted in a greenhouse at the Brazilian Agricultural Research Corporation (Embrapa Maize and Sorghum), located in Sete Lagoas, Minas Gerais, Brazil (19°28′ S, 44°15′08″ W). The biomass sorghum hybrid BRS716 was selected based on its high biomass production potential and suitability for energy cogeneration via biomass combustion, along with its broad adaptability to diverse regions of Brazil.

### 2.1. Melanaphis Sorghi Colony

A colony of *M. sorghi* was maintained on sorghum plants of the cultivar BRS Ponta Negra in 15 L pots, kept in a greenhouse with an average temperature of 26 ± 6 °C and relative humidity of 73%.

### 2.2. Soil Characterization, Preparation, and Fertilization

The soil used in this study was predominantly clayey, composed of 67% clay, 23% silt, and 10% sand. It had a pH of 6.1 (in water) and a silicon content of 12.12 mg kg^−1^. This soil, classified as dystrophic Red Oxisol, was collected from a ravine area at EMBRAPA (Sete Lagoas, Brazil). After collection, the soil was air-dried at an ambient temperature and subsequently sieved through a 2 mm mesh.

The prepared soil was placed in 20 L containers arranged inside a greenhouse. Nine BRS716 sorghum seeds were sown in each container, and after germination, the plants were thinned, maintaining three plants per pot.

Silicon (Si) was manually incorporated into the soil, previously dissolved in water to ensure complete integration. The Si source used was precipitated silicic acid (SiO_2_·xH_2_O, molar mass = 60.08 g mol^−1^; assay ≥ 99% SiO_2_) (Merck KGaA, Darmstadt, Germany) [26]. Sorghum plants were treated with increasing doses of silicic acid (0, 2, 4, or 6 t ha^−1^). Half of each dose was applied at sowing (calculated for a 20 cm soil depth, equivalent to 0.5, 1, or 1.5 g L^−1^, respectively), and the remaining half was applied at the fully expanded five-leaf stage.

The higher doses were selected because sorghum efficiently absorbs Si, allowing us to evaluate the effects of elevated silicic acid concentrations. Additionally, the application timing was synchronized with standard sorghum fertilization practices to ensure agronomic relevance.

For initial fertilization, an 8-28-16 NPK fertilizer (Fertilizantes Heringer, Iguatama, Brazil) was applied at a rate of 80 kg ha^−1^, supplemented with 40 kg ha^−1^ of urea (Fertilizantes Heringer, Iguatama, Brazil), also applied at the fully expanded five-leaf stage.

### 2.3. Assessment of Agronomic Parameters

This experiment was conducted in a greenhouse measuring 12 m (length) × 4 m (width) × 3.20 m (height), covered with light-diffusing polyethylene film, between April and June 2022. Controlled conditions included temperature (26 ± 6 °C, regulated by an automatic exhaust system activated above 30 °C and cooling walls), relative humidity (73 ± 5%, maintained by ventilation and an evaporative system), and natural photoperiod (11.34 ± 0.2 h). Temperature and humidity were monitored using a digital thermometer. A completely randomized design was adopted, with three replicates, each consisting of three plants.

To establish *M. sorghi* colonies on plants (with two-to-three fully expanded leaves), a pot containing *M. sorghi*-infested sorghum plants (cv. BRS Ponta Negra—a cultivar routinely used for rearing aphid colonies under laboratory conditions) was placed equidistantly among every four experimental pots. This method simulated the natural infestation pattern observed in sorghum fields.

Infestation and damage assessments were conducted weekly using a visual observation scale [21,27], adapted from the *M. sacchari* protocol [28] (Table 1 and Table 2; Figure 1 and Figure 2).

Additionally, plant growth characteristics were measured, including height (from soil level to the youngest folded leaf), stem diameter (measured with a digital caliper), and the number of green leaves (not completely dried) per plant.

After harvest, fresh weight was recorded using a digital scale with an accuracy of ±2 g. For dry weight determination, plants were dried in a forced-air oven at 65 °C for 96 h and then weighed on a digital scale (±2 g accuracy).

### 2.4. Chemical and Bromatological Composition Analysis

After drying and weighing, samples were ground in a knife mill and sent to the laboratory for bromatological analysis using near-infrared spectroscopy (NIRS). Samples were placed in glass plates with an internal diameter of 90 mm, used as sample holders for NIR spectra recording. Spectra were collected using a Büchi NIRFlex N-500 FT-NIR spectrometer (Flawil, Switzerland) equipped with a diffuse reflectance accessory. Data were acquired using NIRWare Operator software (version 1.5) and processed in MATLAB (version 7.13) using the PLS Toolbox (version 6.5) PLS routine to determine lignin, cellulose, and hemicellulose contents [29].

Chemical analyses were conducted by the Brazilian Laboratory of Environmental and Agricultural Analyses—LABRAS (Monte Carmelo, Brazil) on dried and ground samples, following the plant nutritional status assessment protocol [30]. Nitrogen and silicon levels were determined after sulfuric acid digestion; phosphorus, potassium, calcium, magnesium, sulfur, copper, iron, manganese, and zinc were quantified after nitroperchloric digestion; and boron was analyzed after dry digestion. These procedures provided a comprehensive characterization of the elements present in plant tissue.

### 2.5. Statistical Analysis

The data were analyzed using analysis of variance (ANOVA) at a significance level of *p* < 0.05, employing the SISVAR statistical software (version 5.8) [31]. When significant effects were detected, linear and quadratic regression analyses were conducted, followed by the Scott–Knott test to compare treatments with and without infestation. Graphs were generated using JAMOVI (version 2.4.11) [32] and Minitab (version 21.1.0) [33].

## 3. Results

### 3.1. Interaction Between Silicon and Melanaphis Sorghi Infestation

Analysis (Appendix A) showed that, at the infestation level, both infestation scores (*μ* = 37.4%) and damage scores (*μ* = 1.40) caused by *M. sorghi* increased over days after emergence (DAE) across all silicon doses evaluated (*p* < 0.01). However, as silicon doses (0, 2, 4, and 6 t ha^−1^) increased, infestation and damage levels in sorghum plants progressively decreased. Plants without silicon application (0 t ha^−1^) exhibited the highest infestation (*μ*Final Score = 98.4%) and damage scores (*μ*Final Score = 4.50), while those treated with the highest dose (6 t ha^−1^) showed the lowest values (*μ*Final Score = 70% and 3.17) (Figure 3).

Plant dry weight (*μ* = 13.15 g) also increased with higher silicon doses (*p* = 0.0036), both in infested (*μ*Dose6 = 14.00 g) and non-infested plants (*μ*Dose6 = 17.06 g) (Figure 4).

Additionally, cellulose concentration (*μ* = 33.64%) was influenced by silicon application (*p* = 0.018). While cellulose levels in non-infested plants remained stable (*μ*Dose6 = 33.47%) with increasing silicon doses, infested plants showed a progressive increase, exceeding 40% at the highest dose (6 t ha^−1^). These findings suggest that silicon may stimulate changes in cell wall composition in response to pest attack (Figure 5).

Manganese (Mn) (*μ* = 71.19 mg/kg) and zinc (Zn) (*μ* = 18.29 mg/kg) concentrations responded differently to silicon doses depending on infestation status. Mn increased in infested plants (*μ*Dose6 = 82.81 mg/kg) with higher silicon doses but decreased in non-infested plants (*μ*Dose6 = 56.14 mg/kg; *p* = 0.001). Conversely, Zn levels decreased in infested plants (*μ*Dose6 = 15.25 mg/kg) with increasing silicon doses but increased in non-infested plants (*μ*Dose6 = 22.17 mg/kg; *p* = 0.008) (Figure 6).

No significant differences were observed in stem diameter (*μ* = 9.60 mm; *p* = 0.134) or the concentrations of sulfur (S) (*μ* = 1.11 g/kg; *p* = 0.269), boron (B) (*μ* = 14.32 mg/kg; *p* = 0.107), copper (Cu) (*μ* = 3.32 mg/kg; *p* = 0.749), or iron (Fe) (*μ* = 143.40 mg/kg; *p* = 0.277) across silicon doses and infestation treatments.

### 3.2. Effect of Silicon on Plant Growth and Composition

Silicon application positively impacted fresh plant weight (*μ* = 94.53 g; *p* = 0.006). A quadratic relationship was observed between silicon doses and final fresh weight (*μ*Dose6 = 109.47 g), with increments up to a saturation point (*μ*Dose4 = 115.72 g), suggesting an optimal dose for maximizing growth (Figure 7).

Regarding cell wall composition, silicon application induced distinct changes in lignin (*μ* = 3.98%; *p* = 0.008) and hemicellulose (*μ* = 28.98%; *p* < 0.001) levels. While lignin decreased (*μ*Dose6 = 3.66%), hemicellulose (*μ*Dose6 = 30.23%) slightly increased, indicating that silicon modulates the biosynthesis and deposition of structural components (Figure 8).

In mineral nutrition, silicon affected macronutrient levels. A 36% reduction in nitrogen (*p* < 0.001) and a 48% reduction in potassium (*p* < 0.001) were observed, while phosphorus (*p* = 0.003) and calcium (*p* < 0.001) increased by 46% and 57%, respectively (Figure 9). Finally, silicon accumulation in plants increased exponentially with applied doses (*p* = 0.012) (Figure 10).

### 3.3. Effect of Melanaphis Sorghi Infestation on Plant Growth and Nutrition

Infestation by *M. sorghi* significantly impaired plant growth and nutritional parameters. In vegetative growth, infested plants exhibited a 17% reduction in average height (*p* = 0.006) compared with non-infested plants (Figure 11A). Consistent results were observed in a 12% reduction in leaf number per plant (*p* = 0.022) (Figure 11B) and a 29% decrease in fresh weight (*p* = 0.002) (Figure 11C).

Concerning structural composition, a 14% reduction in hemicellulose levels (*p* < 0.01) was observed in infested plants (Figure 12A), indicating that infestation negatively altered this cell wall constituent. Infestation also reduced the plants’ calorific value by 3% (*p* = 0.015) (Figure 12B). Regarding mineral nutrition, reductions of 14% in nitrogen (*p* = 0.009) (Figure 13A), 19% in phosphorus (*p* = 0.046) (Figure 13B), and 20% in magnesium (*p* = 0.015) (Figure 13C) were observed in infested plants.

## 4. Discussion

The results of this study demonstrated the impact of silicon (Si) and *M. sorghi* infestation on the growth, mineral nutrition, and structural composition of hybrid biomass sorghum plants (BRS716), providing relevant insights for integrated pest management and nutrient application in agriculture.

Silicon application exhibited a significant protective effect against *M. sorghi*-induced damage, with a marked reduction in infestation and damage levels in plants treated with increasing Si doses. The direct relationship between Si dose and decreased infestation and damage underscores Si’s role in enhancing plant structural integrity. The observed modulation of cellulose concentration supports this hypothesis, as elevated levels of this structural polysaccharide in Si-treated infested plants suggest cell wall reinforcement in response to biotic stress [34,35].

Other changes in cell wall composition included reduced lignin content. Studies indicate that higher Si concentrations may interfere with lignin oligomer aggregation, inhibiting its formation [36]. In tobacco, for instance, Si treatment reduced lignin accumulation even in leaves not directly exposed to mechanical stress, highlighting Si’s modulatory role in stress responses [37]. Additionally, Si stimulates the production of secondary metabolites, such as phenolics, which may influence lignin biosynthesis under stress conditions [38]. This reduction in lignin is particularly advantageous for industrial applications like bioenergy production, as lignin is a major limiting factor in biomass digestibility during biofuel conversion [39].

This selective cell wall reinforcement suggests an adaptive mechanism to biotic stress [40]. Alterations in structural polysaccharides (e.g., cellulose and hemicellulose) increase cell wall rigidity and hinder insect access to cellular contents [41,42]. These structural adaptations not only protect against pest attacks but also enhance the plant’s suitability for industrial applications, such as bioenergy production, underscoring its dual relevance for agriculture and industry [43,44,45,46].

Another contributing factor, widely cited in other studies, is silica deposition in cell walls [47,48]. This physical barrier impedes insect stylet penetration, inhibiting feeding, as described by recent studies [49,50]. Additionally, the Si-induced modulation of chemical compounds may reduce tissue attractiveness to aphids, reinforcing plant defense mechanisms [23].

For instance, Si enhances the activity of defense-related enzymes like peroxidase and polyphenol oxidase, both crucial for plant responses to herbivory [51]. Additionally, Si accumulation in plant tissues has been linked to the increased production of phenolic compounds—natural deterrents against herbivores [52].

Silicon supplementation can also reshape resource allocation in plants, potentially affecting the balance between structural defenses and the biosynthesis of defensive metabolites [53]. Furthermore, Si interacts with plant physiology in complex ways, influencing nutrient uptake and hormonal regulation, which in turn modulate the production of secondary metabolites involved in pest resistance [35]. The Si-driven changes in phenolic content are especially relevant for aphid resistance. These secondary metabolites can disrupt aphid feeding behavior and reproduction [53].

While chewing insects may be more directly affected by structural and chemical barriers, phloem-feeding aphids might experience a less immediate impact [52]. This suggests that Si contributes to a broader and more multi-layered defense strategy, complementing, rather than replacing, the plant’s innate physiological responses to herbivory.

These findings align with prior studies on sap-sucking pest management. Different Si forms (e.g., nano-silica, tetraethyl orthosilicate, Na_2_SiO_3_, and K_2_SiO_3_) reduced aphid density in wheat, with tetraethyl orthosilicate yielding the best results [54]. Higher soil Si levels decreased aphid numbers in both resistant and susceptible wheat varieties, reinforcing Si’s role as a resistance inducer that strengthens cell walls and creates physical barriers against insects [25].

Beyond reducing pest populations, Si also influences aphid life cycle and feeding behavior. A concentration of 2 g·L^−1^ of Si prolonged nymphal duration, reduced longevity and fertility, and decreased feeding preference in *S. avenae* [24]. These behavioral effects stem from Si’s interference with insect feeding and reproduction. Up to 80% mortality in first-instar *S. graminum* nymphs occurred on wheat treated with 100 mL·L^−1^ of Si [55]. 

Other studies corroborate Si’s benefits across crops. Silicic acid treatment reduced *Lipaphis erysimi* (Kalt.) (Hemiptera: Aphididae) populations in rapeseed while improving photosynthesis and stomatal conductance [56]. 

Beyond structural improvements, Si application significantly increases fresh weight and dry biomass, even under *M. sorghi* infestation. This benefit may stem from Si’s ability to enhance photosynthetic efficiency, such as by promoting higher chlorophyll levels (crucial for photosynthesis, as demonstrated in maize under saline stress [57]) and by improving stomatal conductance and transpiration rates, which facilitate gas exchange and CO_2_ uptake [58]. Additionally, Si helps mitigate aphid-induced oxidative stress [59,60]. However, the lower biomass in infested plants indicates that insect damage partially limits growth potential, likely due to continuous sap extraction, which compromises water and nutrient uptake [8,61,62]. These results emphasize the need for complementary strategies to maximize Si’s benefits.

From a nutritional perspective, Si application altered macronutrient levels, reducing nitrogen and potassium concentrations. This may reflect competition between Si and these nutrients for specific transporters [63,64,65]. Conversely, increased phosphorus and calcium levels suggest Si selectively modulates nutrient absorption and redistribution, preserving those critical for energy metabolism and structural integrity [65,66,67,68].

The modulation of manganese and zinc concentrations in this study may reflect complex interactions among Si, nutrients, and biotic stress [69]. Future research should explore how these interactions influence plant resistance to pests and fertilizer-use efficiency. While Si offers clear benefits, its management must be optimized to avoid adverse effects on plant nutrition.

Stem diameter and sulfur (S), iron (Fe), copper (Cu), and boron (B) concentrations were unaffected by Si or *M. sorghi* infestation. The lack of significant changes in stem diameter suggests this structural trait is less sensitive to Si or aphid presence compared with other growth and compositional parameters. For micronutrients (Fe, Cu, B) and S, stable concentrations may indicate a limited interaction with Si or a minimal impact of infestation on their uptake and redistribution. This stability likely reflects the plant’s ability to maintain the homeostasis of these elements under biotic stress, given their roles in essential metabolic processes like photosynthesis and protein synthesis [70].

Infestation by *M. sorghi* negatively impacted growth, structural composition, and nutrient levels, reducing nitrogen, phosphorus, and magnesium concentrations. Declines in height, leaf number, and fresh weight highlight the deleterious effects of aphid sap consumption, which impairs nutrient transport and photosynthesis. Similar studies reported substantial damage in *M. sorghi*-infested crops, including reduced vegetative vigor and nutrient-use efficiency [14,15]. Aphid-induced stress disrupts nutrient absorption and redistribution [71]. Moreover, reduced hemicellulose and nitrogen levels in infested plants support the hypothesis that aphid attacks compromise both metabolic functionality and mechanical resistance, adversely affecting agricultural productivity [72,73].

Infestation also lowered the plants’ calorific value, likely due to reduced hemicellulose content and biomass loss from aphid feeding. This decline may impair the biomass’s energy efficiency for cogeneration, underscoring the need for management strategies to mitigate aphid damage.

The interaction between Si and *M. sorghi* in this study revealed Si’s potential to partially offset infestation-related damage, positioning it as a tool for integrated pest management. The 6 t ha^−1^ Si dose was most effective, minimizing infestation severity and promoting the highest dry weight increase under biotic stress. However, the agronomic viability of this dose requires careful evaluation, considering application costs, Si source availability, and potential impacts on nutrient uptake. 

Because Si is non-essential and does not fully eliminate *M. sorghi* damage, its economic justification for large-scale use may be limited. Nevertheless, in some contexts, Si application offers measurable financial benefits. For example, foliar Si spraying (1.50 mL/L) in Indian sorghum fields increased yields by 2850 kg/ha, achieving a favorable benefit/cost ratio of 1.61 [74]. Economic returns, however, depend on stress severity, yield potential, and crop variety. While Si can mitigate losses under moderate-to-high stress, its benefits may not outweigh costs in low-yielding systems or less responsive cultivars [75,76].

Therefore, integrated approaches—combining reduced Si doses with complementary management practices—could offer a more sustainable and cost-effective pest control strategy. Similar to the use of natural enemies [77], silicon supplementation may improve the suppression of parasitoids on *M. sorghi*, highlighting its potential for integrated pest management (IPM) in sorghum [78]. These synergistic effects could enhance agricultural resilience and ecosystem health. Future studies should investigate such combinatorial strategies to optimize Si’s role in crop protection and bioenergy production.

## 5. Conclusions

Infestation by *M. sorghi* negatively impacts plant growth, nutrient uptake, and structural integrity. However, Si application mitigates these adverse effects by enhancing plant metabolism and resistance.

The addition of Si increases the resistance of biomass sorghum BRS716 to *M. sorghi,* significantly reducing infestation levels and damage. Increasing Si doses improves both fresh and dry biomass accumulation and modifies cell wall composition by elevating cellulose content in infested plants.

Plant-accumulated Si levels rise exponentially with applied doses, confirming Si as a promising tool for the sustainable management of biomass sorghum BRS716. This enhances biotic stress tolerance and agronomic performance.

## Figures and Tables

**Figure 1 insects-16-00566-f001:**
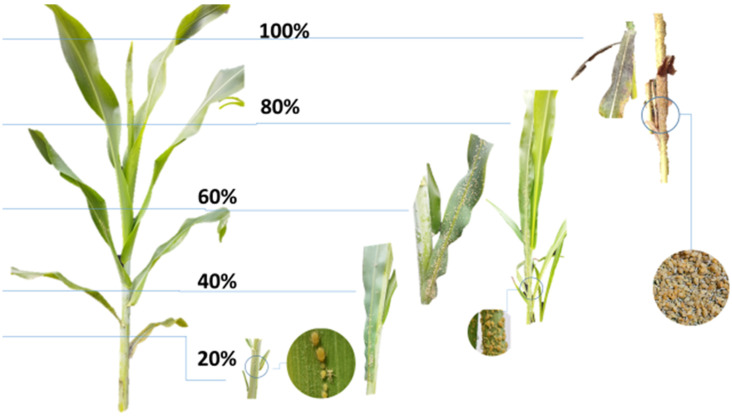
Infestation scale for evaluation of sorghum aphid (*Melanaphis sorghi*).

**Figure 2 insects-16-00566-f002:**
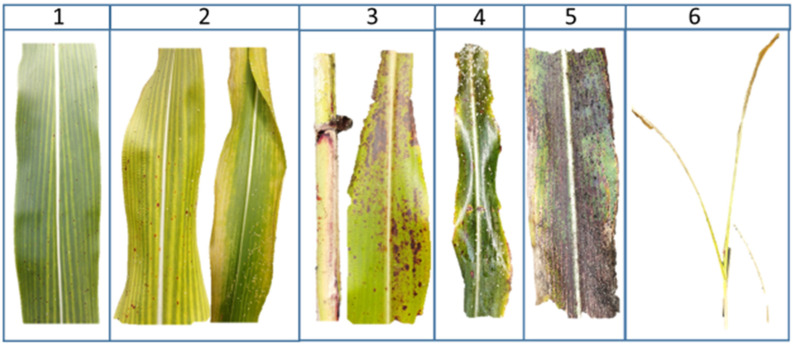
Damage scale for evaluation of sorghum aphid (*Melanaphis sorghi*). Score 1: Few visible lesions; Score 2: Initial reddish spots on the midrib and presence of exuviae; Score 3: Scattered reddish spots, leaf mar-gin yellowing, and moderate exuviae; Score 4: Leaves with reddish and yellowish spots, bronzed appearance, and abundant exuviae; Score 5: Leaves exhibiting reddish-yellow discoloration, initial leaf tissue desiccation, and reduced exuviae; Score 6: Complete plant death.

**Figure 3 insects-16-00566-f003:**
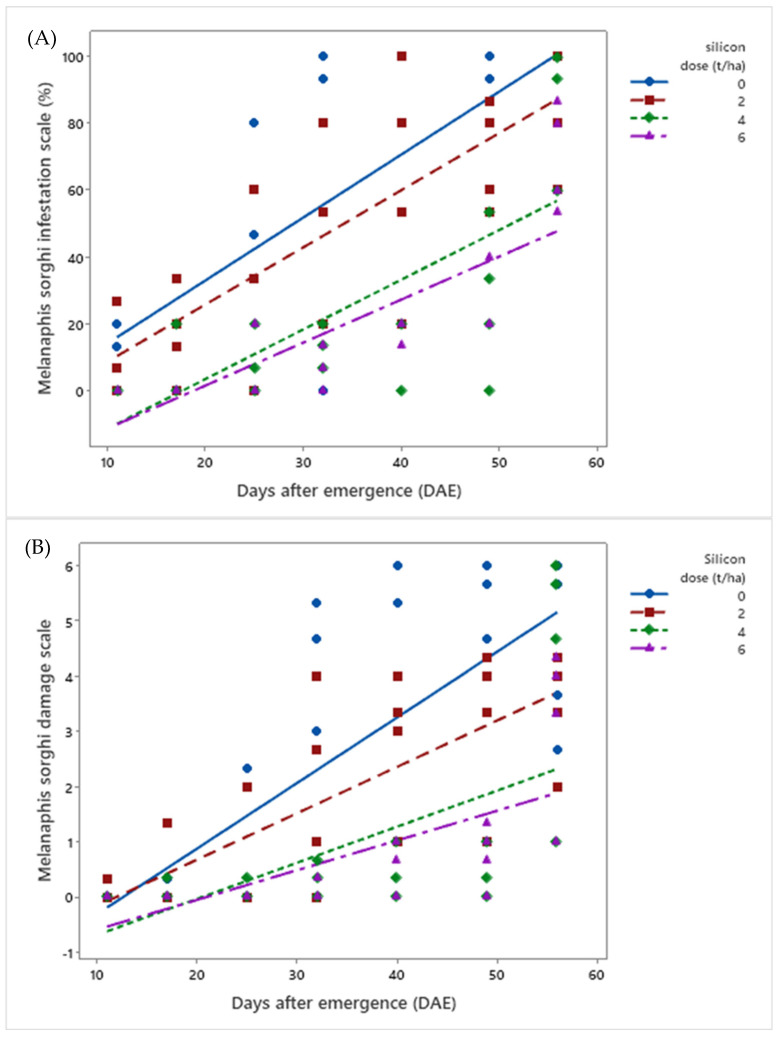
Linear regressions for (**A**) infestation score and (**B**) damage score of *Melanaphis sorghi* in biomass sorghum subjected to four silicic acid doses over 56 days after emergence. Regression equations for infestation: Dose 0: y = 0.106x − 0.433 (R^2^ = 93.58%); Dose 2: y = 0.086x − 0.428 (R^2^ = 95.66%); Dose 4: y = 0.080x − 1.46 (R^2^ = 63.16%); Dose 6: y = 0.060x − 1.22 (R^2^ = 73.42%). Regression equations for damage: Dose 0: y = 0.119x − 1.500 (R^2^ = 88.42%); Dose 2: y = 0.084x − 1.011 (R^2^ = 93.87%); Dose 4: y = 0.065x − 1.351 (R^2^ = 47.50%); Dose 6: y = 0.053x − 1.139 (R^2^ = 74.63%). Location: Sete Lagoas, Minas Gerais, Brazil, 2025.

**Figure 4 insects-16-00566-f004:**
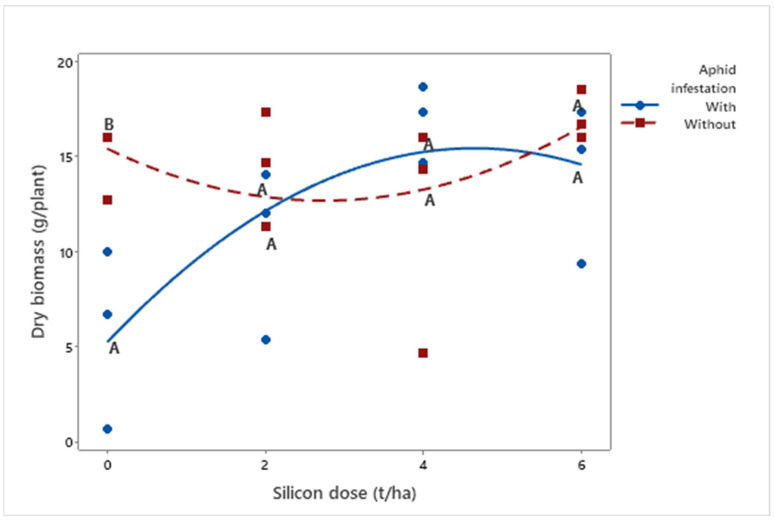
Quadratic regressions of dry biomass (g/plant) as a function of silicon dose (t/ha) in biomass sorghum plants with and without *Melanaphis sorghi* infestation. Different letters denote significant differences (*p* < 0.05) between with and without-infested treatments. The regression equations were as follows: infested plants: y = −0.472x^2^ + 4.388x + 5.224 (R^2^ = 91.02%); non-infested plants: y = 0.364x^2^ − 2.001x + 15.414 (R^2^ = 62.59%). Location: Sete Lagoas, Minas Gerais, Brazil, 2025.

**Figure 5 insects-16-00566-f005:**
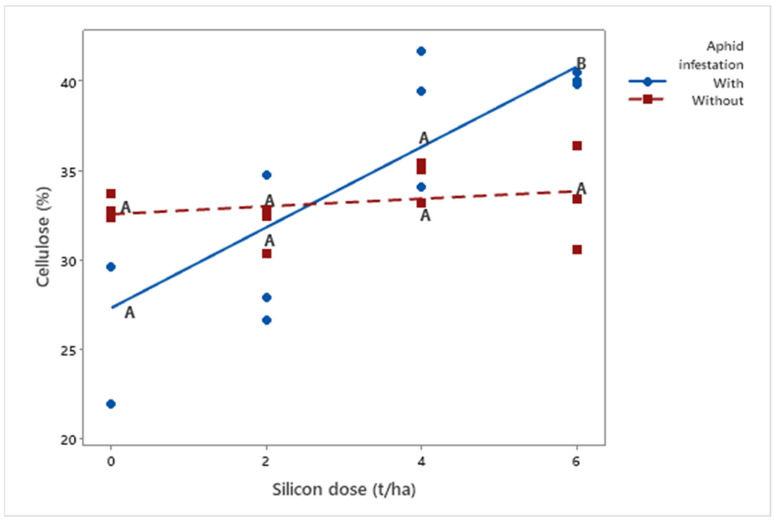
Regression analysis of cellulose percentage as a function of silicon dose (t/ha) in biomass sorghum plants with and without *Melanaphis sorghi* infestation. Different letters denote significant differences (*p* < 0.05) between with and without -infested treatments. The regression equations were as follows: infested plants: y = −2.246x + 27.33 (R^2^ = 91.29%); non-infested plants: y = 0.212x + 32.574 (R^2^ = 24.51%). Location: Sete Lagoas, Minas Gerais, Brazil, 2025.

**Figure 6 insects-16-00566-f006:**
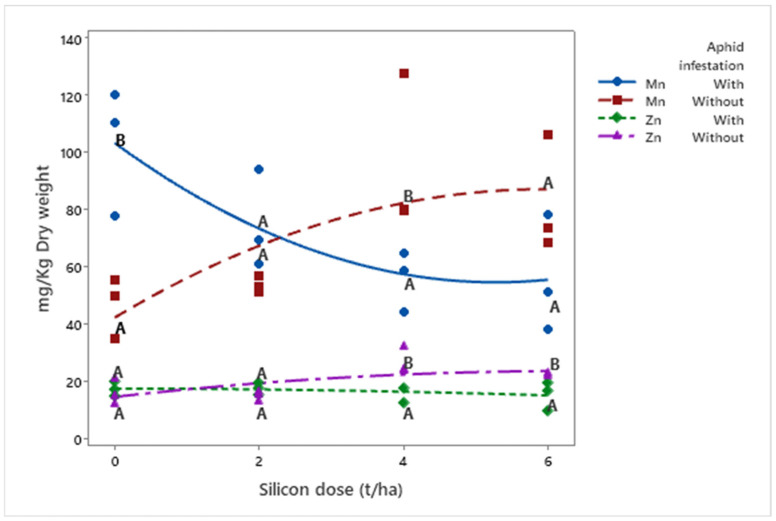
Concentrations of Mn and Zn (mg kg^−1^ dry weight) in biomass sorghum plants subjected to different silicon doses (t ha^−1^), with and without *Melanaphis sorghi* infestation. Solid lines represent trends for Mn under infestation and Zn under infestation, while dashed lines represent trends without infestation. Different letters denote significant differences (*p* < 0.05) between with and without -infested treatments. Regression equations for manganese: with infestation: y = 1.748x^2^ − 18.433x + 103.330 (R^2^ = 99.66%); without infestation: y = −1.260x^2^ + 15.053x + 12.361 (R^2^ = 75.10%). Regression equations for zinc: with infestation: y = −0.062x^2^ − 0.032x + 17.536 (R^2^ = 91.63%); without infestation: y = −0.222x^2^ + 2.838x + 14.600 (R^2^ = 75.10%). Location: Sete Lagoas, Minas Gerais, Brazil, 2025.

**Figure 7 insects-16-00566-f007:**
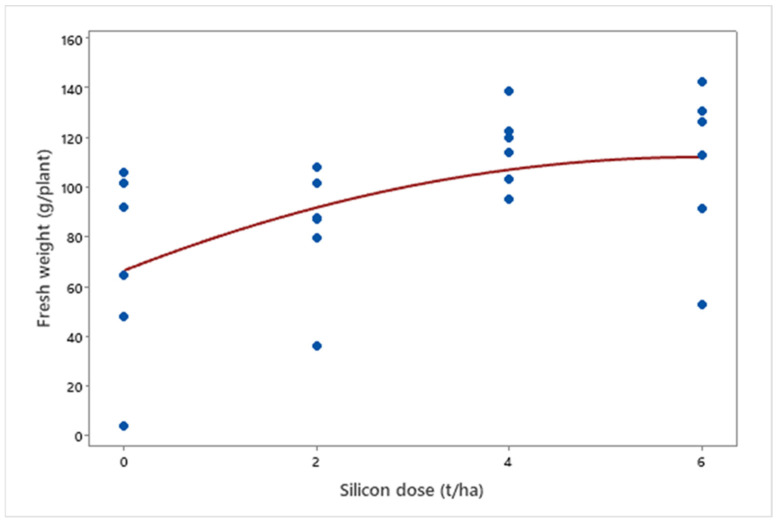
Fresh weight of biomass sorghum plants treated with different Si doses (0, 2, 4, and 6 t ha^−1^) at 56 days after emergence. Regression equation: y = −1.262x^2^ + 15.183x + 66.657 (R^2^ = 88.63%). Location: Sete Lagoas, Minas Gerais, Brazil, 2025.

**Figure 8 insects-16-00566-f008:**
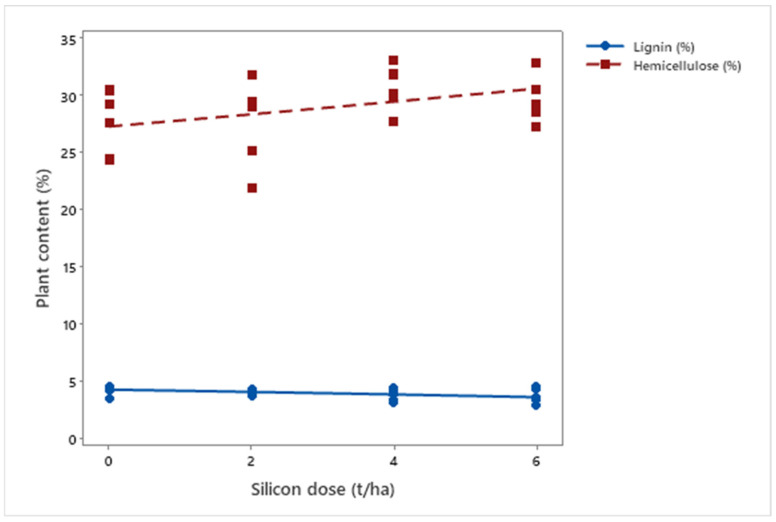
Lignin and hemicellulose concentrations in biomass sorghum plants subjected to different silicon doses (t ha^−1^). Regression equation: hemicellulose: y = 0.553x + 27.318 (R^2^ = 62.30%).

**Figure 9 insects-16-00566-f009:**
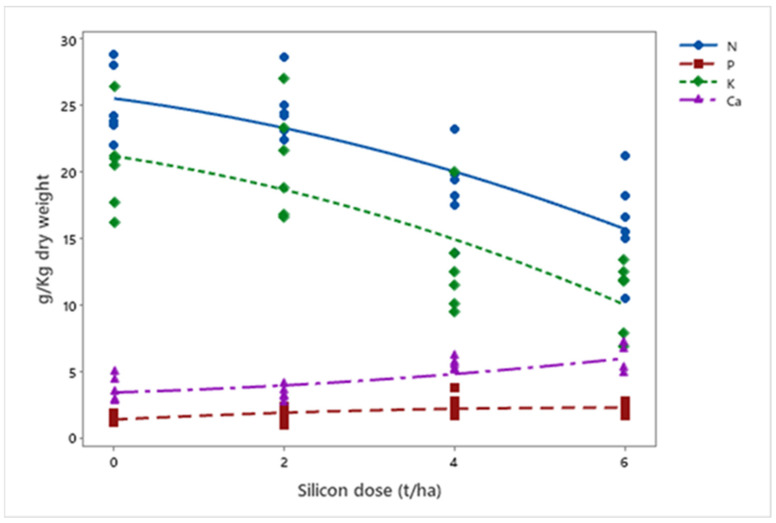
Concentrations of N, P, K, and Ca (g kg^−1^ dry weight) in biomass sorghum plants subjected to different silicon doses (t ha^−1^). Regression equations: N: y = −0.130x^2^ − 0.852x + 25.500 (R^2^ = 93.08%); P: y = −0.027x^2^ + 0.311x + 1.347 (R^2^ = 69.52%); K: y = −0.149x^2^ − 0.965x + 21.188 (R^2^ = 88.43%); Ca: y = 0.040x^2^ + 0.185x + 3.390 (R^2^ = 77.18%). Location: Sete Lagoas, MG, Brazil, 2025.

**Figure 10 insects-16-00566-f010:**
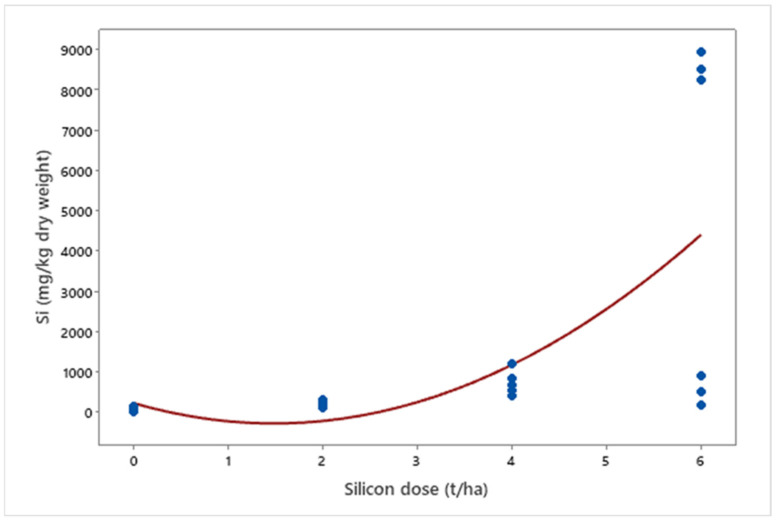
Silicon (Si) concentrations (mg kg^−1^ dry weight) in biomass sorghum plants subjected to different silicon doses (t ha^−1^). Regression equation: y = 229.605x^2^ − 680.835x + 223.529 (R^2^ = 97.11%). Location: Sete Lagoas, MG, Brazil, 2025.

**Figure 11 insects-16-00566-f011:**
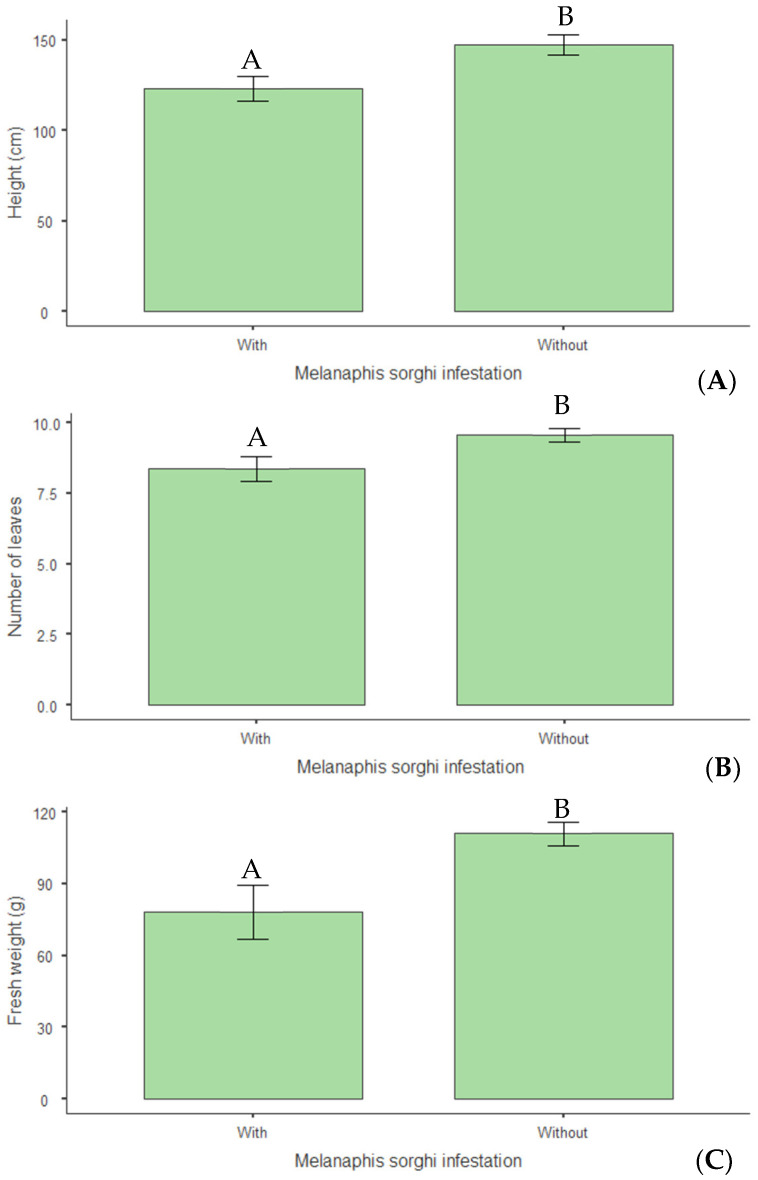
(**A**) Plant height (cm); (**B**) number of leaves; (**C**) fresh weight (g) per plant of biomass sorghum subjected to different silicon doses (t ha^−1^), with and without *Melanaphis sorghi* infestation. Different letters denote significant differences (*p* < 0.05) between with and without -infested treatments. Location: Sete Lagoas, MG, Brazil, 2025.

**Figure 12 insects-16-00566-f012:**
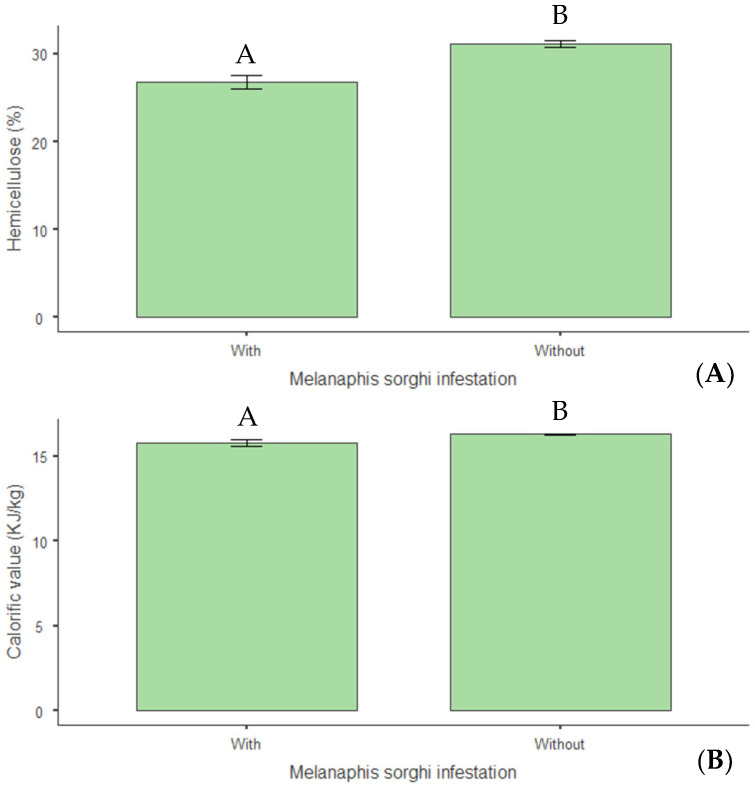
(**A**) Hemicellulose (%) and (**B**) calorific value (kJ kg^−1^) in biomass sorghum plants subjected to different silicon doses (t ha^−1^), with and without *Melanaphis sorghi* infestation. Different letters denote significant differences (*p* < 0.05) between with and without -infested treatments. Location: Sete Lagoas, MG, Brazil, 2024.

**Figure 13 insects-16-00566-f013:**
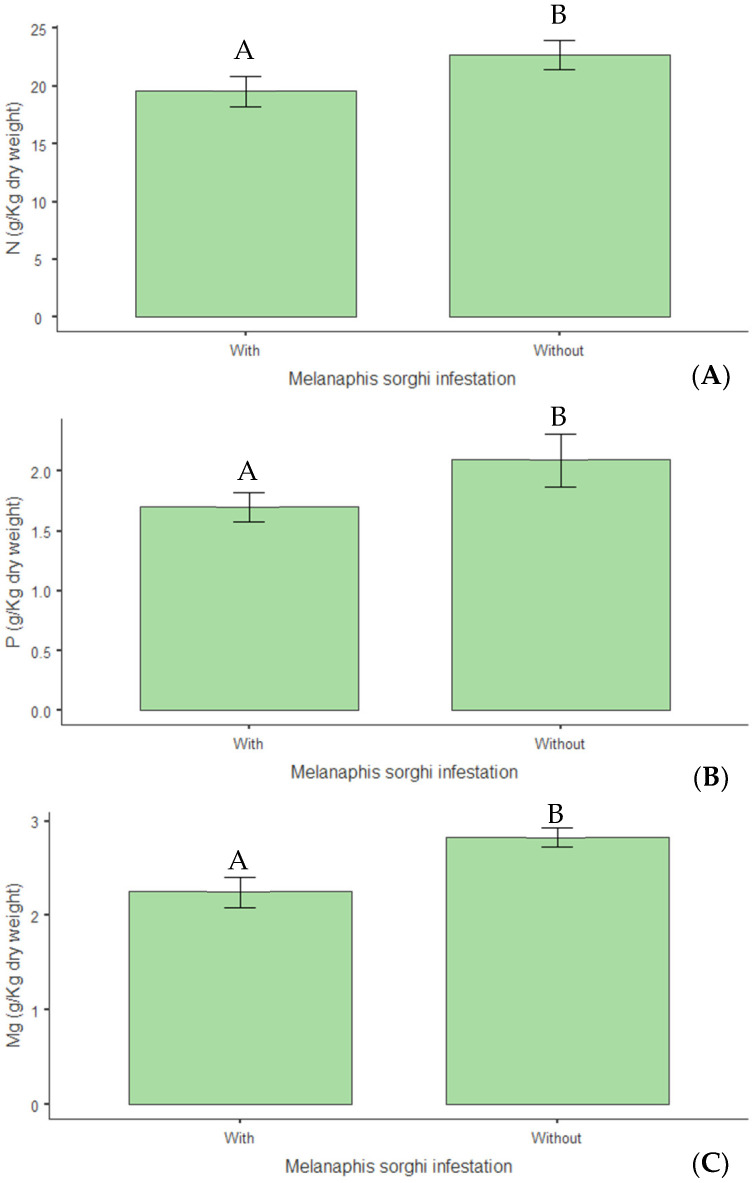
(**A**) Nitrogen (N); (**B**) phosphorus (P); (**C**) magnesium (Mg) content (g kg^−1^ dry weight) in biomass sorghum subjected to different silicon doses (t ha^−1^), with and without *Melanaphis sorghi* infestation. Different letters denote significant differences (*p* < 0.05) between with and without -infested treatments. Location: Sete Lagoas, MG, Brazil, 2025.

**Table 1 insects-16-00566-t001:** Infestation scale of *Melanaphis sorghi* in sorghum.

Infestation Level	Aphid Density	Distribution	Symptoms
20%	<10 aphids/leaf	Lower leaves and stem	No exuviae
40%	<50 aphids/leaf	Lower leaves, stem, and middle leaves	Scattered exuviae
60%	>50 aphids/leaf	Lower and middle leaves dominated	Abundant exuviae; initial yellowing/reddening
80%	>200 aphids/leaf	Central veins and leaf margins affected	Evident symptoms; high exuviae density
100%	Complete colonization	Entire plant	Leaf necrosis; generalized symptoms

**Table 2 insects-16-00566-t002:** Damage severity scale of *Melanaphis sorghi* in sorghum.

Score	Symptoms	Visual Indicators
1	Few visible lesions	Minor discoloration
2	Initial reddish spots on the midrib; presence of exuviae	Localized damage and chlorosis
3	Scattered reddish spots; leaf margin yellowing; moderate exuviae	Chlorosis; exuviae accumulation and reddish spots
4	Reddish-yellowish spots; bronzed appearance; abundant exuviae	Advanced discoloration; exuviae density and honeydew accumulation
5	Reddish-yellow discoloration; initial leaf desiccation; reduced exuviae	Severe tissue damage; early necrosis and sooty mold fungus
6	Complete plant death	Total collapse of plant structure

## Data Availability

The original contributions presented in this study are included in the article/Appendix A. Further inquiries can be directed to the corresponding author.

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
