# Peer review of "Biomass Sorghum (Sorghum bicolor) Agronomic Response to Melanaphis sorghi (Hemiptera: Aphididae) Infestation and Silicon Application"

_insects, 2025, doi:10.3390/insects16060566_

Round 1
Reviewer 1 Report
Comments and Suggestions for Authors
Line 40: What does high dry weight mean?. Please mention the percentage relative to the negative control for greater precision.
Line 64: To support this statement, please cite the following reference: A Review of Biological Control One Decade After the Sorghum Aphid (Melanaphis sorghi) Outbreak. https://doi.org/10.3390/plants13202873
Line 105 to 107: It is mentioned that each pot contained three plants. However, in lines 118–119, it is stated that the treatments consisted of four plants per replicate. This discrepancy is unclear: how were four plants used per replicate if each pot only held three plants?. Clarification of this section in the Materials and Methods is necessary.
Line 138 and 145: The citation must be presented with a number according to the journal's format.
Line 176: The results presented in this sentence for the treatment of 6 t·ha⁻¹ (damage: 3.17; infestation: 70%) are inconsistent with Figure 3. According to Figure 3B, M. sorghi infestation reached approximately 40%, while Figure 3A shows damage levels of around 1.5. I recommend reviewing the results presented throughout the manuscript for accuracy.
The Y-axis in Figure 8 has no units.
Figure 9 should appear after its first mention in the manuscript, not before.
In Section 3.2, "Effect of Silicon on Plant Growth and Composition," it is mentioned that hemicellulose levels slightly increase with silicon concentration. However, in Section 3.3, "Effect of Melanaphis sorghi Infestation on Plant Growth and Nutrition," a reduction in hemicellulose levels is observed. Subsequently, changes in nitrogen concentration and the other macroelements mentioned above are monitored once again. The way the results are presented may be confusing to the reader. I recommend displaying the results more clearly.
The bars in Figures 11, 12, and 13 do not show statistical analysis.
Figure 16 does not exist; there is an error in the figure number. It should be Figure 13.
Line 501: To support your claims at the end of this paragraph, please cite the following reference: A Review of Biological Control One Decade After the Sorghum Aphid (Melanaphis sorghi) Outbreak. https://doi.org/10.3390/plants13202873
Reviewer 2 Report
Comments and Suggestions for Authors
Please refer to and carefully address the detailed comments and suggestions provided in the attached review report. Following these points will help improve the clarity, rigor, and overall quality of the manuscript.

Round 2
Reviewer 1 Report
Comments and Suggestions for Authors
The article is now ready for publication.